# Association of *TNF*-α 308G/A and *LEPR* Gln223Arg Polymorphisms with the Risk of Type 2 Diabetes Mellitus

**DOI:** 10.3390/genes13010059

**Published:** 2021-12-27

**Authors:** Maria Trapali, Dimitra Houhoula, Anthimia Batrinou, Anastasia Kanellou, Irini F. Strati, Argyris Siatelis, Panagiotis Halvatsiotis

**Affiliations:** 1Department of Biomedical Medicine, University of West Attica, 12243 Attiki, Greece; ytrapali@uniwa.gr; 2Department of Food Science and Technology, University of West Attica, 12243 Attiki, Greece; dhouhoula@uniwa.gr (D.H.); batrinou@uniwa.gr (A.B.); akanellou@uniwa.gr (A.K.); estrati@uniwa.gr (I.F.S.); 33rd Department of Urologist Medical School, National and Kapodistrian University of Athens, “ATTIKON” University Hospital, 12462 Chaidari, Greece; argysiat@yahoo.gr; 42nd Propaedeutic Department of Internal Medicine, Medical School, National and Kapodistrian University of Athens, “ATTIKON” University Hospital, 12462 Chaidari, Greece

**Keywords:** *TNF*-α- 308G/A, type 2 diabetes mellitus, polymorphism, leptin receptor gene (*LEPR*) Gln223Arg

## Abstract

The objective of the present study was to identify the association of the *TNF*-α- 308G/A and leptin receptor (*LEPR*) Gln223Arg polymorphisms with the risk of development of type 2 diabetes mellitus (T2DM). Methods: A total of 160 volunteers were studied: 108 with T2DM and 52 participants as control, who served as the control group. Polymerase chain reaction–restriction fragment length polymorphism (PCR-RFLP) for the genomic region of *TNF*-α- 308G/A and *LEPR* Gln223Arg were carried out. Results: The frequency of *LEPR* Gln223Arg genotypes in T2DM and control groups showed significant differences in the distribution of genotypes (*p* < 0.05). The frequency also of *TNF-α-* 308G/A genotypes in T2DM and control subjects showed significant differences in the distribution of genotypes (*p* < 0.05). Conclusion: Our results indicate that there are significant differences in the distribution of genotypes and alleles between the individuals with T2DM and control subjects (*p* < 0.05).

## 1. Introduction

Type 2 diabetes mellitus (T2DM) is characterized as a disorder of impaired energy metabolism caused by insufficient insulin production and/or decreased insulin sensitivity. The etiology remains unclear but many environmental and polygenic factors may contribute to the development of T2DM. Although the pathogenesis of T2DM is still unclear, the increasing prevalence of the disorder is of great importance and a global public health concern. HbA_1_c is the gold standard for evaluating the long-term glycaemic control in diabetes, because it accurately reflects real glycaemic levels in vivo [1] The American Diabetes Association (ADA) treatment guidelines suggest an HbA_1_c level <7.0% as the primary glycaemic control target for people with diabetes, and the decrease in HbA_1_c level consequently reduces the prevalence of the devastating diabetes-related chronic complications [2].

Adipose tissue is a very active endocrine organ, secreting a number of hormones, such as adiponectin, leptin, resistin and visfatin, collectively with classical cytokines, such as tumor necrosis factor-alpha (*TNF*-α) and interleukin-6 (IL-6). All these adipocytokines play significant role in the regulation of energy metabolism, the metabolism of glucose and lipids, reproduction, cardiovascular function and immunity [3]. *TNF*-α plays a critical role in several autoimmune diseases, such as rheumatoid arthritis, and may mediate obesity-linked insulin resistance in type 2 diabetes. Upregulated expression of *TNF*-α plays a significant role in the induction of insulin resistance linked with obesity and T2DM [4]. Genetic variations in the promoter region of *TNF-α* may regulate transcription and production of the protein and may influence inflammatory diseases. Various studies have investigated single nucleotide polymorphisms (SNPs) in the promoter region of the human *TNF* gene, such as 238G/A, 308G/A, 857C/T and 1031T/C [5]. Two studies relate the studied polymorphism in gene 308G/A with T2DM and coronary heart disease, while in two other studies, their presence was related in people with type 2 diabetes, burdened with nephropathy. Nevertheless, there are studies that relate directly *TNF* gene polymorphism with T2DM development [6]. Chronic low-grade inflammation, which has been referred as “metaflammation”, is considered as a relevant factor contributing to the development of diabetic complications [7]. However, the association between the *TNFα* promoter genotypes and the risk of T2DM development remains controversial [8].

Leptin is an adipocytokine, considered as a crucial metabolic regulator with a primary role in reducing food intake, increasing energy expenditure and to regulating immune and inflammatory responses by binding and activating the leptin receptor [9,10]. Several studies relate leptin gene (*LEP*) and leptin receptor gene (*LEPR*) polymorphisms with obesity [11,12], insulin resistance [13] and T2DM [14]. The gene polymorphisms of *LEP* G2548A (rs7799039: guanine > adenine) and *LEPR* Gln223Arg (rs1137101: glutamine > arginine) have been examined extensively and significant associations of *LEPR* Gln223Arg polymorphism, where the circulating leptin and glucose levels were observed [15]. A mutation in the exon 6 of *LEPR* “rs1137101 (Gln223Arg)”, has been extensively studied. This specific polymorphism has been reported in several populations. It has an A to G transition, in which a codon CAG changes to CGG, resulting in the substitution of glutamine by arginine (Gln223Arg) in the *LEPR*. The location of this mutation is on the 6th exon of the gene and may impair signal transduction enough to increase susceptibility to T2DM. This study aimed to compare the two gene polymorphisms 308G/A (*TNF-α*) and Gln223Arg (*LEPR*) with the risk of T2DM development in a cohort of people with T2DM in the Greek population, for which limited data exist.

## 2. Materials and Methods

### 2.1. Collection of Clinical Samples

A total of 160 venous blood samples were collected from 108 volunteers (58 men and 50 women) with T2DM with a mean age of 67.5 ± 12.6 years old. A group of 52 volunteers (22 men and 30 women) without diabetes or other chronic diseases with a mean age of 51 ± 17.7 years old served as the control sample. Blood was collected in an EDTA containing vacutainer tube early in the morning and then transported in sterile tubes and stored immediately thereafter at −20 °C. Plasma glucose was measured using local laboratories with blood collected at the time of presentation to the hospital and the patients were not necessarily fasting prior to blood collection. All participants resided in the municipal area of Athens and the study period was from early June to late July 2021. All volunteers with GHbA1c > to 7.5% were considered as people with type 2 diabetes mellitus, while GHbA1c levels with normal fasting blood sugar levels and without any abnormal measurements in the past history were accepted as non-diabetic participants.

### 2.2. Questionnaire

We evaluated the demographics data (sex and age) and biochemical parameters (glucose, glucosylated heamoglobin). 

### 2.3. Genomic DNA Extraction 

DNA was extracted directly from blood specimens using an automatic extractor with the Whole Blood Nucleic Acid Extraction Kit (ZYBIO Company, Germany) following the protocol recommended by the supplier. The purity and the quantity of extracted DNA were evaluated spectrophotometrically by calculating OD260/OD280 (spectrophotometer Epoch, Biotek, Winusky, VT, USA).

### 2.4. Polymerase Chain Reaction–Restricted Fragment Length Polymorphism (PCR-RFLP) for the Genomic Region of TNF-α -308G/A

After checking for the purity of DNA, PCR was carried out. The chosen primers amplify a size of 107 bp the genomic region of *TNF-α*- 308G/A single nucleotide polymorphism in the promoter region of *TNF* using 0.3 μL of primers forward 5′-AGGCAATAGGTTTTGAGGGCCAT-3′ and reverse 5′-TCCTCCCTGCTCCGATTCCG-3′. PCR was performed in 50 μL final volume solution using the Master Mix (PCRBIO TaqMix Red, PCRBIOSYSTEMS, London, UK). The amplification was conducted by a thermal cycler (96-well thermal cycler Applied Biosystems, Waltham, MA, USA), as follows: an initial denaturation: 95 °C, 3 min; 40 cycles with the following step-cycle profile: denaturation 95 °C, 15 s; annealing 60 °C, 15 s; extension 72 °C, 60 s; final extension 72 °C, 10 min. 

PCR products were separated in 2% agarose gel, stained with ethidium bromide (0.5 μg/mL) and documented under UV illumination using MiniBIS Pro device (DNR Bio-Imaging Systems Ltd., Neve Yamin, Israel). 

The 107 bp PCR product was digested with *NcoI* (New England Biolabs, London, UK) restriction enzyme for 15 min at 37 °C. Three types of bands were observed—a complete *NcoI* cut representing homozygous *TNF-α* (-308G/G), resulting in two fragments of 87 and 20 bp; a partial cut representing heterozygous *TNF-α* (-308G/A), resulting in three fragments of 107, 87 and 20 bp; and an uncut 107 bp fragment representing homozygous *TNF-α* (-308A/A).

### 2.5. Polymerase Chain Reaction–Restricted Fragment Length Polymorphism (PCR-RFLP) for the Genomic Region of LEPR Gln223Arg

After checking for the purity of DNA, PCR was carried out. The chosen primers amplify a size of 421 bp the genomic region of *LEPR* Gln223Arg using 0.3 μL of the following primers: forward primer—5′-ACCCTTTAAGCTGGGTGTCCCAAATAG-3′; reverse primer—5′-AGCTAGCAAATATTTTTGTAAGCAATT-3′. PCR was performed in 50 μL final volume solution using Master Mix (PCRBIO TaqMix Red). The program of the PCR was as follows: an initial denaturation: 95 °C, 3 min; 40 cycles with the following step-cycle profile: denaturation 95 °C, 15 s; annealing 60 °C, 15 s; extension 72 °C, 60 s; final extension 72 °C, 10 min. PCR products were separated in 2% agarose gel, stained with ethidium bromide (0.5 μg/mL) and documented under UV illumination using a MiniBIS Pro device (DNR Bio-Imaging Systems Ltd., Neve Yamin, Israel). 

The 421 bp PCR product was digested with *MspI* (Thermo Scientific, Waltham, MA, USA) restriction enzyme for 16 h at 65 °C. Three types of bands were observed—a complete MspI cut representing homozygous *LEPR* Gln223Arg (Arg/Arg), resulting in two fragments of 294 and 127 bp; a partial cut representing heterozygous Gln/Arg (421, 294 and 127 bp), resulting in three fragments of 421, 294 and 127 bp; and an uncut 421 bp fragment representing homozygous Gln/Gln (Figure 1).

### 2.6. Biomedical Ethics Issues

Institutional ethical approval was received prior to the study, and the collection of all epidemiological data were conducted in such a way as to fully guarantee and preserve anonymity and confidentiality. All the questionnaires were collected with the consent of the volunteers. 

### 2.7. Statistical Analysis

The null hypothesis is accepted, since *p* > 0.05. The *p* value can be calculated in Excel (Microsoft, Redmond, Wash) with the formula chi-square distribution (CHISQ.DIST, X; df; T), where *X* is the value of the χ^2^-test, *df* is degrees of freedom and *T(RUE)* corresponds to the cumulative χ^2^ distribution. To find the correct value at the χ^2^ distribution table, we need the degrees of freedom, which are calculated by the following formula: df = (r − 1) × (c − 1), where *c* is the number of columns and *r* is the number of rows in our 2 × 2 table. The df = (2 − 1) × (2 − 1) = 1. The allele frequency among all participants is in accordance to the Hardy–Weinberg equilibrium. Multinomial logistic regression analysis was performed by SPSS software (Version 26).

## 3. Results

Overall, 50% of the volunteers were male and 50% were female. Out of these 160 participants, 67.5% (*n* = 108) had T2DM. The median glucose on blood collection time was 150 mg/dL for people with T2DM, while in 5.6% (*n* = 6) of them, a glucose level of ≥ 200 mg/dL was recorded. A glucose level < 85 mg/dL characterized all participants of the control group. All volunteers with diabetes presented with HbA1c > 7.0%. 

Genotyping of the *LEPR* Gln223Arg and the genomic region of *TNF-α* -308G/A was assessed in the 52 controls and 108 T2DM participants. The frequency of *LEPR* Gln223Arg genotypes in T2DM and control groups is shown in Table 1, with significant differences in the distribution of genotypes (*p* < 0.05). 

Individuals with diabetes who were homozygous on the A and G alleles were 42.6% and 13%, respectively, while heterozygous were 44.4%. The control group, which was homozygous on the G and A allele, was 7.7 and 65.4%, respectively. Statistically significant differences were also detected between the different groups (*p* < 0.05). In contrast, heterozygotes in the control group accounted only for 26.9%. All T2DM volunteers who were homozygous on allele A had plasma glucose levels between 124–304 mg/dL.

As for the Gln223Arg polymorphism (G-A), allele G occurs with a frequency of 35.2% in T2DM and 21.2% in controls. The A allele occurs with a frequency of 64.8% in T2DM and 78.8% in the control group (Table 2). 

Genotyping of the *TNF-α* -308G/A was assessed in 52 control subjects and 108 T2DM volunteers. The frequency of *TNF-α* -308G/A genotypes in T2DM and control subjects are shown in Table 3, with significant differences in the distribution of genotypes (*p* < 0.05). People with diabetes who were homozygous on the A and G alleles were 0% and 68.5%, respectively, while T2DM subjects who were heterozygous were 31.5%. In the control group homozygous in G was the 57.7% while none (0%) was homozygous for the A allele. In contrast, heterozygotes in the control group accounted for 42.5%. 

For the *TNF-α* -308G/A polymorphism the (G-A) form, it was shown that allele A occurs with a frequency of 15.7% in T2DM and 24% in controls. The G allele occurs with a frequency of 84.3% in T2DM and 76% in the control group (Table 4). Finally, all the 13 T2DM who were homozygous in the G allele, for the *LEPR*Gln223Arg polymorphism, were homozygous on the G allele of the *TNF-α* -308G/A polymorphism. Table 5 presents the frequency of genotypes for both polymorphisms studied among genders, revealing non statistically significant differences (*p* > 0.05) in the T2DM group between the two genders except for the GA genotype of *LEPR* Gln223Arg.

Multinomial logistic regression analysis was performed for the *LEPR* and *TNF*-a genotypes and the results are presented in Table 6, Table 7, Table 8, Table 9, Table 10 and Table 11, respectively. The independent variables chosen were the following: gender (two categories: male, female), age group (three categories: 0–45, 46–65, >65) and T2DM group (two categories: diabetic, normal or control).

In Table 6, from “Model Fitting Information”, it is observed that the full model statistically significantly (*p* = 0.001) predicts the dependent variable (*LEPR* genotypes) better than the intercept-only model alone. In Table 7, “Likelihood Ratio Tests” present the independent variables that are statistically significant (i.e., age group variable and T2DM group variable with *p* = 0.006 and *p* = 0.000, respectively). In Table 8, “Parameter Estimates”, the first set of coefficients represents comparisons between AA and GG genotypes. “Age group >65” and “T2DM Diabetic group” were significant predictors: (b = 1.618, s.e. = 0.692, *p* = 0.019) and (b = −2.343, s.e. = 0.787, *p* = 0.003), respectively. Individuals scoring higher on the variable “T2DM Diabetic group” were less likely to have AA genotype. The odds ratio of 0.096 indicates that for every one unit increase on “T2DM Diabetic group” the odds of an individual having AA genotype changed by a factor of 0.096 (in other words, the odds were decreasing). The second and final set of coefficients represents comparisons between GA and GG genotypes. “Age group 0–45” and “T2DM Diabetic group” were negative predictors, although the specific parameters were not significant to the model.

In Table 9, from “Model Fitting Information”, it is observed that the model was not statistically significantly improved (*p* = 0.220), compared to the intercept-only model alone. “Likelihood Ratio Tests” (Table 10) showed that none of the independent variables (gender, age group and T2DM group) are statistically significant (*p* > 0.05). In Table 11, “Parameter Estimates”, the coefficients represent comparisons between GA and GG genotypes. More specifically, “age group >65”, “age group 0–45” and “T2DM Diabetic group” were negative predictors, although the specific parameters were not significant (*p* > 0.05) to the model.

## 4. Discussion

In our study, we reported an association between the leptin receptor gene Q223R polymorphism and the *TNF-α* -308G/A in a sample of T2DM volunteers. T2DM is considered a disease caused by a combination of genes and this polygenic nature has been linked to obesity, hypertension, gout, and disordered lipid metabolism. Leptin is a hormone derived from fat that may play a role in regulating metabolism of energy and body lipid homeostasis. The receptor of this hormone, the *LEPR*, belongs to the superfamily of cytokine receptors and is a single transmembrane protein with many spliced isoforms (one long isoform and several short isoforms) distributed in several tissues. The long isoform of *LEPR* is biologically active and abundantly expressed in the hypothalamus, where it activates the Janus kinase signal transducer and the activation of transcription (STAT) system resulting in alterations of transcription or translation of hypothalamic neuropeptides. Single nucleotide mutations (SNP’s) of the leptin receptor (*LEPR*) gene that result in a premature termination of the intracellular domain of the protein have been considered responsible for obesity. In particular, the Q223R polymorphisms (glutamine to arginine at codon 223) result in substitutions of amino acids in the extracellular region of the *LEPR* that may have potential functional consequences [12,15]. In the current analysis, a significant association between the *LEPR* Gln223Arg gene polymorphism and T2DM in the sample population was observed. Our analyses also revealed significant associations between *TNF-α* -308G/A and T2DM. Tumor necrosis factor-α (TNF-α) is a cytokine that has been found to play a role in insulin resistance and the *TNF-α* gene is one of the candidate genes implicated in type 2 diabetes mellitus (T2DM).

Our findings are in line with some studies regarding *LEPR* Gln223Arg expression [16], whereas they are contradictory to others; for example, [17,18] concluded that *LEPR* Gln223Arg gene polymorphism had no effect on the susceptibility to T2DM and did not consider any differences in regard of race and ethnicity. However, studies that examine different ethnic groups may yield conflicting results. *LEPR* Gln223Arg gene polymorphism has been associated with an increased risk of T2DM and thus, carriers of the G allele of *LEPR* Gln223Arg gene polymorphism may be more susceptible to T2DM than non-carriers of the G allele. However, our study is in agreement with a meta-analysis conducted in 2016 that has examined 11 studies (~5000 cases in total) and has concluded that the rs1137101(G) allele is associated significantly with T2DM, with an odds ratio of about 1.2–1.8 [19,20,21].

Based on the literature, it is inconclusive whether the -308G/A polymorphisms in the *TNF-α* promoter results in T2DM susceptibility [22,23,24,25,26]. However, our findings follow recent meta-analysis studies [27,28], suggesting that *TNF-α* -308G/A can be considered as a risk factor for T2DM.

A leading factor that may contribute to the development of T2DM is the composition of nutrients that an individual consumes, such as carbohydrates, fats, proteins, minerals and vitamins, which play a vital role in metabolic homeostasis. [29]. Additionally, gut microbiota alterations have been found not only in obesity but in multiple pathologies, ranging from intestinal bowel syndrome (IBS) and type 2 diabetes (T2D) [30]. Intervention studies are needed to explore the role of applied nutrigenetic and precision nutrition to counteract T2DM [31]. In this context, precision nutrition with customized dietary recommendations may be applied to an individual, in order to prevent and manage chronic diseases. Moreover, data suggest that the methylation levels of *LEPR* and the promoter of *TNF-α* could be used as epigenetic biomarkers to respond to a low-calorie diet. In addition, the methylation profile of the promoters of these genes could help predict susceptibility to comorbidities, such as hypertension or type 2 diabetes [32]. Future larger studies should be considered necessary, investigating different age groups as well. Therefore, while the main objective of our study was to investigate certain polymorphisms, we could not reproduce our conclusions for the complete genome variations of the studied biomarkers. In the future, it could be possible to schedule personalized nutritional strategies, with corrective dietary actions for the particular negative SNP carriers (*LEPR* Gln223Arg and *TNF-α* -308G/A). These solutions, eventually, may contribute to a better management of T2DM.

## 5. Conclusions

Our study concludes that the G allele in the *LEPR* Gln223Arg and *TNF-α* -308G/A polymorphisms has a significant higher frequency in T2DM. However, the limitation of our study is that the sample size is small and we are lacking an evaluation of an extensive number of possibly significant demographic and biochemical factors, that may play a role in the plasma *LEPR* and *TNF* levels. More research is therefore required to verify our conclusion.

## Figures and Tables

**Figure 1 genes-13-00059-f001:**
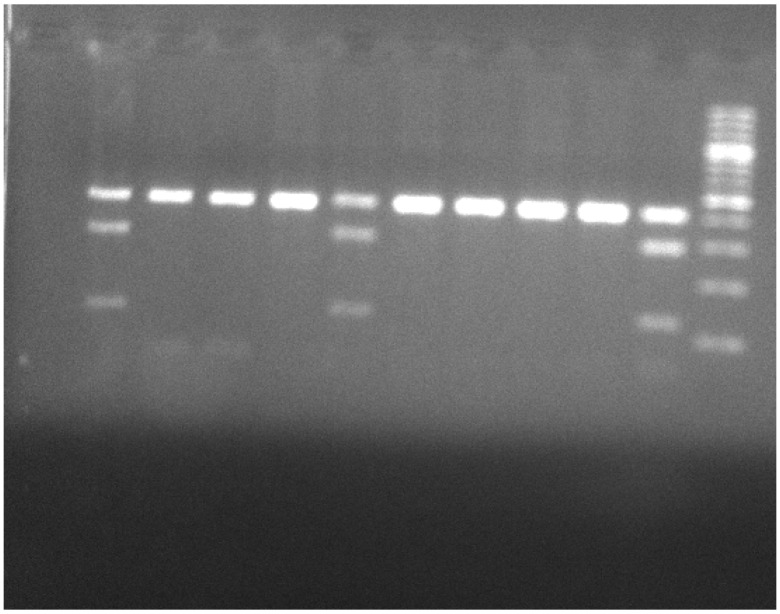
Heterozygous G/A samples are represented in lanes 2, 6 and 11, in which the *TNF-α* promoter gene was partially cut with the *MspI*, resulting in 3 DNA fragments of 440, 300 and 140 bp, respectively. Homozygous A/A samples are represented by lanes 3–5 and 7–10 in which an uncut 440 bp fragment is observed. Lane 1 is a negative control and Lane 12, the DNA Ladder, is 100 bp.

**Table 1 genes-13-00059-t001:** Frequency of *LEPR* Gln223Arg genotypes in T2DM and control groups.

*LEPR* Gln223Arg	T2DM	Control
Gln/Gln (AA)	46 (42.6%) ^a^	34 (65.4%) ^b^
Gln/Arg (AG)	48 (44.4%) ^a^	14 (26.9%) ^c^
Arg/Arg (GG)	14 (13%) ^b^	4 (7.7%) ^a^

Notes: ^a,b,c^—different superscript letters in the same column or row indicate statistically significant differences (*p* < 0.05).

**Table 2 genes-13-00059-t002:** Frequency of *LEPR* Gln223Arg alleles in T2DM and control groups.

*LEPR* Gln223Arg	T2DM	Control
G	35.2% ^a^	21.2% ^b^
A	64.8% ^b^	78.8% ^a^

Notes: ^a,b^—different superscript letters in the same column or row indicate statistically significant differences (*p* < 0.05).

**Table 3 genes-13-00059-t003:** Frequency of *TNF-α* -308G/A genotypes in T2DM and control subjects.

*TNF-α* -308G/A	T2DM	Control
G/G	74 (68.5%) ^a^	27 (51.9%) ^b^
G/A	34 (31.5%) ^c^	25 (48.1%) ^b^

Notes: ^a,b,c^—different superscript letters in the same column or row indicate statistically significant differences (*p* < 0.05).

**Table 4 genes-13-00059-t004:** Frequency of *TNF-α* -308G/A alleles in T2DM and control groups.

*TNF-α* -308G/A	T2DM	Control
G	84.3% ^a^	76% ^b^
A	15.7% ^b^	24.0% ^a^

Notes: ^a,b^ Different superscript letters in the same column or row indicate statistically significant differences (*p* < 0.05).

**Table 5 genes-13-00059-t005:** Frequency of *TNF-α* -308G/A and *LEPR* Gln223Arg genotypes in T2DM and control subjects.

*Gender*	*TNF-α* -308G/A (GA)	*TNF-α* -308G/A (GG)	*LEPR* Gln223Arg GG	*LEPR* Gln223Arg GA	*LEPR* Gln223Arg AA
Female (Control Group)	17 (56.7%) ^a^	13 (43.3%) ^a^	3 (10%) ^a^	8 (26.7%) ^b^	19 (63.3%) ^c^
Male (Control Group)	8 (36.4%) ^b^	14(63.6%) ^c^	1 (4.5%) ^a^	6 (27.3%) ^b^	15 (68.2%) ^c^
Female T2DM	17(34%) ^b^	33 (66%) ^c^	4 (8%) ^a^	28 (56%) ^c^	18 (36%) ^b^
Male T2DM	17 (29.3%) ^b^	41(70.7%) ^c^	10 (17.2%) ^a^	20 (34.5%) ^b^	28 (48.3%) ^b^

Notes: ^a,b,c^—different superscript letters in the same column or row (for each polymorphism) indicate statistically significant differences (*p* < 0.05).

**Table 6 genes-13-00059-t006:** Multinomial logistic regression analysis for the *LEPR* Gln223Arg genotypes. Model Fitting Information.

Model	Model Fitting Criteria	Likelihood Ratio Tests		
	−2 Log Likelihood	Chi-Square	df	Sig.
Intercept Only	85.541			
Final	60.156	25.385	8	0.001

Sig. stands for “significance probability”.

**Table 7 genes-13-00059-t007:** Multinomial logistic regression analysis for the *LEPR* Gln223Arg genotypes. Likelihood Ratio Tests.

Effect	Model Fitting Criteria	Likelihood Ratio Tests		
	−2 Log Likelihood of Reduced Model	Chi-Square	df	Sig.
Intercept	60.156 ^a^	0.000	0	
gender	64.007	3.851	2	0.146
agegroup	74.445	14.289	4	0.006
T2DM	76.766	16.610	2	0.000

The chi-square statistic is the difference in −2 log-likelihoods between the final model and a reduced model. The reduced model is formed by omitting an effect from the final model. The null hypothesis is that all parameters of that effect are 0. ^a^ This reduced model is equivalent to the final model because omitting the effect does not increase the degrees of freedom. Sig. stands for “significance probability”.

**Table 8 genes-13-00059-t008:** Multinomial logistic regression analysis for the *LEPR* Gln223Arg genotypes. Parameter Estimates.

*LEPR* ^a^		B	Std. Error	Wald	df	Sig.	Exp(B)	95% Confidence Interval for Exp(B)	
								Lower Bound	Upper Bound
AA	Intercept	2.756	0.808	11.650	1	0.001			
	[gender = FEMALE]	0.263	0.575	0.209	1	0.648	1.300	0.421	4.013
	[gender = MALE]	0 ^b^	.	.	0	.	.	.	.
	[agegroup = >65]	1.618	0.692	5.465	1	0.019	5.043	1.299	19.580
	[agegroup = 0–45]	−1.535	0.790	3.774	1	0.052	0.215	0.046	1.014
	[agegroup = 45–65]	0 ^b^	.	.	0	.	.	.	.
	[T2DM = DIABETIC]	−2.343	0.787	8.853	1	0.003	0.096	0.021	0.449
	[T2DM = NORMAL]	0 ^b^	.	.	0	.	.	.	.
GA	Intercept	1.382	0.827	2.791	1	0.095			
	[gender = FEMALE]	0.853	0.574	2.211	1	0.137	2.347	0.762	7.222
	[gender = MALE]	0 ^b^	.	.	0	.	.	.	.
	[agegroup = >65]	1.015	0.686	2.188	1	0.139	2.760	0.719	10.590
	[agegroup = 0–45]	−1.072	0.767	1.955	1	0.162	0.342	0.076	1.538
	[agegroup = 45–65]	0 ^b^	.	.	0	.	.	.	.
	[T2DM = DIABETIC]	−0.832	0.781	1.135	1	0.287	0.435	0.094	2.011
	[T2DM = NORMAL]	0 ^b^	.	.	0	.	.	.	.

^a^ The reference category is: GG. ^b^ This parameter is set to zero because it is redundant. Sig. stands for “significance probability”. Std. Error: Standard Error.

**Table 9 genes-13-00059-t009:** Multinomial logistic regression analysis for the *TNF-a -*308G/A genotypes. Model Fitting Information.

Model	Model Fitting Criteria	Likelihood Ratio Tests		
	−2 Log Likelihood	Chi-Square	df	Sig.
Intercept Only	42.626			
Final	36.887	5.739	4	0.220

**Table 10 genes-13-00059-t010:** Multinomial logistic regression analysis for the *TNF-a -*308G/A genotypes. Likelihood Ratio Tests.

Effect	Model Fitting Criteria	Likelihood Ratio Tests		
	−2 Log Likelihood of Reduced Model	Chi-Square	df	Sig.
Intercept	36.887 ^a^	0.000	0	.
gender	38.162	1.275	1	0.259
agegroup	39.169	2.283	2	0.319
T2DM	39.962	3.075	1	0.080

The chi-square statistic is the difference in −2 log-likelihoods between the final model and a reduced model. The reduced model is formed by omitting an effect from the final model. The null hypothesis is that all parameters of that effect are 0. ^a^ This reduced model is equivalent to the final model because omitting the effect does not increase the degrees of freedom.

**Table 11 genes-13-00059-t011:** Multinomial logistic regression analysis for the *TNF-a -*308G/A genotypes. Parameter Estimates.

*TNF* ^a^		B	Std. Error	Wald	df	Sig.	Exp(B)	95% Confidence Interval for Exp(B)	
								Lower Bound	Upper Bound
GA	Intercept	−0.124	0.403	0.094	1	0.759			
	[gender = FEMALE]	0.383	0.340	1.268	1	0.260	1.466	0.753	2.855
	[gender = MALE]	0 ^b^	.	.	0	.	.	.	.
	[agegroup = >65]	−0.112	0.392	0.082	1	0.775	0.894	0.414	1.928
	[agegroup = 0–45]	−0.764	0.519	2.173	1	0.140	0.466	0.168	1.286
	[agegroup = 45–65]	0 ^b^	.	.	0	.	.	.	.
	[T2DM = DIABETIC]	−0.726	0.417	3.032	1	0.082	0.484	0.214	1.095
	[T2DM = NORMAL]	0 ^b^	.	.	0	.	.	.	.

^a^ The reference category is: GG. ^b^ This parameter is set to zero because it is redundant.

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
