# Peer review of "Association of TNF-α 308G/A and LEPR Gln223Arg Polymorphisms with the Risk of Type 2 Diabetes Mellitus"

_genes, 2021, doi:10.3390/genes13010059_

Round 1

Reviewer 1 Report

The authors aimed to identify the association of the TNF-alpha 308G/A and Leptin Receptor (LEPR) Gln223Arg polymorphisms with the risk of development of type 2 diabetes mellitus (T2DM).

The study covers some issues that have been overlooked in other similar topics. The structure of the manuscript appears adequate and well divided in the sections. The methodology is well described with enough experimental data and results to support the work. The manuscript needs moderate grammar correction. Please also check typos thorough the text.

Other issues: Limitations of the study should be better reported in the discussion, as well the role of microbiota in obesity (please see and briefly discuss: doi: 10.3390/biology9120415). Conclusions need to be increased with a sentence on future insights on this matter.

Author Response

Response to Reviewer 1 comments

Point 1: The authors aimed to identify the association of the TNF-alpha 308G/A and Leptin Receptor (LEPR) Gln223Arg polymorphisms with the risk of development of type 2 diabetes mellitus (T2DM). The study covers some issues that have been overlooked in other similar topics. The structure of the manuscript appears adequate and well divided in the sections. The methodology is well described with enough experimental data and results to support the work. The manuscript needs moderate grammar correction. Please also check typos thorough the text.

Response 1: Authors would like to thank the Reviewer for his valuable comments regarding the structure and the methodology of the manuscript. Grammar correction was performed as well as typos were checked throughout the manuscript.

Point 2: Other issues: Limitations of the study should be better reported in the discussion, as well the role of microbiota in obesity (please see and briefly discuss: DOI: 10.3390/biology9120415). Conclusions need to be increased with a sentence on future insights on this matter.

Response 2: Limitations of the study were further reported in the Discussion section as well as the role of gut microbiota in obesity (Lines 267-281), according to the Reviewer’s suggestion. Additionally, suggestions for further research were also discussed in the Conclusions.

Reviewer 2 Report

The authors have presented and designed the experiments well.

Just a minor thought, could authors see the gender difference, if so can

include a table for the same.

Author Response

Response to Reviewer 2 comments

Point 1: The authors have presented and designed the experiments well. Just a minor thought, could authors see the gender difference, if so can include a table for the same.

Response 1: Authors included Table 5, according to the Reviewer’s suggestion.

Reviewer 3 Report

In the present study, Trapali and colleagues investigated the association between TNF-15alpha 308G/A (rs1800629) and Leptin Receptor (LEPR) Gln223Arg (rs1137101) polymorphisms and T2DM. Their study included 160 subjects (52 controls and 108 T2DM). Their analysis found a significant difference in allele frequencies of the two polymorphisms between the control and T2DM groups.

Comments:

  1. Between lines 31 and 46, there are numerous statements that could usefully be supported by references.
  2. During sample collection, who was considered a T2DM patient and what were the criteria for being a control.
  3. The sentences between lines 151 and 155 fit into the Introduction, as they are not part of the findings.
  4. The p-value for the results presented in Table 2 is sufficient to be reported once, as it is the result of one test (this is also true for Table 4).
  5. From the results in Table 3, it appears that no person carries the A/A alleles. How could the HW equilibrium have been achieved in this way in the study?
  6. The style of the references is not uniform.

Suggestions:

The statistical analysis used in this study is limited to the application of Chi-squared test. In any case, it would be worthwhile to carry out more complex analyses.

  1. A case-control approach to analysis allows the use of logistic regression. It would be worthwhile to adjust for age, gender, and other known confounders.
  2. A linear regression is also possible based on fasting blood glucose and HbA1C levels.
  3. It may also be interesting to investigate the distribution of polymorphisms by multiple inheritance (dominant or recessive). It is possible that it is different from the commonly used codominant one.
  4. It would also be worthwhile to investigate the combined effect of the two polymorphisms. Chi-square tests and linear and logistic regression can be used for this analysis.

A weakness of the manuscript, besides the limited number of statistical tests used, is the extent to which the results can be considered novel, as the association of both polymorphisms with T2DM has been described previously in the literature:

  • Liu, Z.H.; Ding, Y.L.; Xiu, L.C.; Pan, H.Y.; Liang, Y.; Zhong, S.Q.; Liu, W.W.; Rao, S.Q.; Kong, D.L. A meta-analysis of the association between TNF-alpha -308G>A polymorphism and type 2 diabetes mellitus in Han Chinese population. PLoS One 2013, 8, e59421, doi:10.1371/journal.pone.0059421.
  • Shi, L.X.; Zhang, L.; Zhang, D.L.; Zhou, J.P.; Jiang, X.J.; Jin, Y.L.; Chang, W.W. Association between TNF-alpha G-308A (rs1800629) polymorphism and susceptibility to chronic periodontitis and type 2 diabetes mellitus: A meta-analysis. J Periodontal Res 2021, 56, 226-235, doi:10.1111/jre.12820.
  • Shoily, S.S.; Ahsan, T.; Fatema, K.; Sajib, A.A. Common genetic variants and pathways in diabetes and associated complications and vulnerability of populations with different ethnic origins. Sci Rep 2021, 11, 7504, doi:10.1038/s41598-021-86801-2.
  • Li, Y.Y.; Wang, H.; Yang, X.X.; Wu, J.J.; Geng, H.Y.; Kim, H.J.; Yang, Z.J.; Wang, L.S. LEPR gene Gln223Arg polymorphism and type 2 diabetes mellitus: a meta-analysis of 3,367 subjects. Oncotarget 2017, 8, 61927-61934, doi:10.18632/oncotarget.18720.

Author Response

Response to Reviewer 3 comments

In the present study, Trapali and colleagues investigated the association between TNF-15alpha 308G/A (rs1800629) and Leptin Receptor (LEPR) Gln223Arg (rs1137101) polymorphisms and T2DM. Their study included 160 subjects (52 controls and 108 T2DM). Their analysis found a significant difference in allele frequencies of the two polymorphisms between the control and T2DM groups.

Comments:

Point 1: Between lines 31 and 46, there are numerous statements that could usefully be supported by references.

Response 1: References were inserted in lines 31 and 46, as suggested by the Reviewer.

Point 2: During sample collection, who was considered a T2DM patient and what were the criteria for being a control.

Response 2: All volunteers with GHbA1c > to 7.5% were considered as people with type 2 diabetes mellitus, while GHbA1c levels with normal fasting blood sugar levels and without any abnormal measurements in the past history, were accepted as non-diabetic participants. The relative sentence was inserted in the Materials and Methods section (Lines 95-98).

Point 3: The sentences between lines 151 and 155 fit into the Introduction, as they are not part of the findings.

Response 3: Authors agree with the Reviewer’s suggestion and the sentences were moved in the Introduction (Lines 75-81).

Point 4: The p-value for the results presented in Table 2 is sufficient to be reported once, as it is the result of one test (this is also true for Table 4).

Response 4: Tables 1-4 were revised, according to the Reviewer’s suggestion.

Point 5: From the results in Table 3, it appears that no person carries the A/A alleles. How could the HW equilibrium have been achieved in this way in the study?

Response 5:

It has been reported in similar studies that the allele A in the promoter region of the TNF-α gene at position −308 is a rare one and the genotype A/A cannot be easily detected. In the study of Dedoussis et al. (2005), that studied the association between TNF-α −308G>A polymorphism and the development of acute coronary syndromes in Greek subjects, the genotype frequencies were patients, 87% (n = 206), 12% (n = 29), and 1% (n = 2) for G/G, G/A, and A/A, and in controls, 96% (n = 227), 4% (n = 10), and 0% (n = 0) for G/G, G/A, and A/A, respectively (p = 0.04).

Dedoussis, G., Panagiotakos, D., Vidra, N. et al. Association between TNF-α −308G>A polymorphism and the development of acute coronary syndromes in Greek subjects: The CARDIO2000-GENE Study. Genet Med 7, 411–416 (2005).

Point 6: The style of the references is not uniform.

Response 6: References were revised according to the Reviewer’s suggestion.

Point 7:

Suggestions:

The statistical analysis used in this study is limited to the application of the Chi-squared test. In any case, it would be worthwhile to carry out more complex analyses.

  1. A case-control approach to analysis allows the use of logistic regression. It would be worthwhile to adjust for age, gender, and other known confounders.
  2. Linear regression is also possible based on fasting blood glucose and HbA1C levels.
  3. It may also be interesting to investigate the distribution of polymorphisms by multiple inheritances (dominant or recessive). It is possible that it is different from the commonly used codominant one.
  4. It would also be worthwhile to investigate the combined effect of the two polymorphisms. Chi-square tests and linear and logistic regression can be used for this analysis.

Response 7:

The authors consider that the Chi-squared test is adequate to illustrate the frequencies of genotypes in the T2DM group and control group for the specific polymorphisms studied, which was our initial aim. More complex statistical analyses were not conducted because our aim was not to correlate biochemical parameters (e.g. glucose levels, etc.) with genotypes of specific polymorphisms, but to investigate how the specific SNPs (LEPR Gln223Arg and TNF-α−308G/A) are associated with T2DM in the Greek population, for which limited data exist. Additionally, according to the recent bibliography (Shoily et al., 2021), this information builds up to an existing valuable data base of appropriate biomarkers for effective future therapies.

Point 8:

A weakness of the manuscript, besides the limited number of statistical tests used, is the extent to which the results can be considered novel, as the association of both polymorphisms with T2DM, has been described previously in the literature:

  • Liu, Z.H.; Ding, Y.L.; Xiu, L.C.; Pan, H.Y.; Liang, Y.; Zhong, S.Q.; Liu, W.W.; Rao, S.Q.; Kong, D.L. A meta-analysis of the association between TNF-alpha -308G>A polymorphism and type 2 diabetes mellitus in Han Chinese population. PLoS One 2013, 8, e59421, doi:10.1371/journal.pone.0059421.
  • Shi, L.X.; Zhang, L.; Zhang, D.L.; Zhou, J.P.; Jiang, X.J.; Jin, Y.L.; Chang, W.W. Association between TNF-alpha G-308A (rs1800629) polymorphism and susceptibility to chronic periodontitis and type 2 diabetes mellitus: A meta-analysis. J Periodontal Res 2021, 56, 226-235, doi:10.1111/jre.12820.
  • Shoily, S.S.; Ahsan, T.; Fatema, K.; Sajib, A.A. Common genetic variants and pathways in diabetes and associated complications and vulnerability of populations with different ethnic origins. Sci Rep 2021, 11, 7504, doi:10.1038/s41598-021-86801-2.
  • Li, Y.Y.; Wang, H.; Yang, X.X.; Wu, J.J.; Geng, H.Y.; Kim, H.J.; Yang, Z.J.; Wang, L.S. LEPR gene Gln223Arg polymorphism and type 2 diabetes mellitus: a meta-analysis of 3,367 subjects. Oncotarget 2017, 8, 61927-61934, doi:10.18632/oncotarget.18720.

Response 8: Authors believe that data regarding the association between the studied polymorphism TNF-alpha G-308A with the various chronic diseases are conflicting and not well understood. Moreover, these associations in general, seem to vary from country to country as it is pointed out in the study of Shoily et al, 2021, therefore it is important to investigate the frequencies of polymorphisms in specific ethnic groups and geographic areas. In this context, the study that was conducted on Caucasian Greek individuals is novel. The study of Shoily et al. (2021) was added to the discussion.

Round 2

Reviewer 3 Report

I accept the Authors' answers to points 1 to 6. However, their response to my suggestions is not acceptable to me.

Response 7:

The authors consider that the Chi-squared test is adequate to illustrate the frequencies of genotypes in the T2DM group and control group for the specific polymorphisms studied, which was our initial aim.

  • I respect the Authors' original aim, which was to compare allele frequencies between case and control groups. The Chi2 test used to test this is consistent with this aim. The CHi2 test has limited statistical power and does not correct for potential confounders.

More complex statistical analyses were not conducted because our aim was not to correlate biochemical parameters (e.g. glucose levels, etc.) with genotypes of specific polymorphisms, but to investigate how the specific SNPs (LEPR Gln223Arg and TNF-α−308G/A) are associated with T2DM in the Greek population, for which limited data exist.

  • If the aim was to investigate the association between T2DM and the two SNPs studied, I particularly do not understand why the logistic regression analysis I suggested is not performed. This statistical analysis could confirm that, beyond the difference in allele frequencies, there is an age- and sex-independent (and any other covariate used) association between the SNPs and T2DM.
  • Furthermore, if it is possible to investigate the association between SNPs and T2DM (and the biochemical parameters associated with it) using more complex statistical methods, the new results of which would confirm the previous one obtained from the Chi2 test already presented, then I really do not understand why these analyses are not being done.
  • The small sample size (n = 160) and non-random sampling (case-control) would also justify the use of additional statistical methods.

„Additionally, according to the recent bibliography (Shoily et al., 2021), this information builds up to an existing valuable data base of appropriate biomarkers for effective future therapies.”

  • In that article, they used the 1000 Genomes phase 3 release database, which was designed to create a global reference database (ref: Auton, Adam, Gonçalo R. Abecasis, David M. Altshuler, et al. "A Global Reference for Human Genetic Variation." Nature, vol. 526, no. 7571, 2015, pp. 68-74.). The 160 samples collected in the present study would not be considered representative of the general Greek population (which was not the aim of the present study), but therefore not applicable to perform analyses similar to Shoily et al.

Response 8:

Authors believe that data regarding the association between the studied polymorphism TNF-alpha G-308A with the various chronic diseases are conflicting and not well understood. Moreover, these associations in general, seem to vary from country to country as it is pointed out in the study of Shoily et al, 2021, therefore it is important to investigate the frequencies of polymorphisms in specific ethnic groups and geographic areas. In this context, the study that was conducted on Caucasian Greek individuals is novel. The study of Shoily et al. (2021) was added to the discussion.

  • I agree with the Authors that genetic analysis should be carried out on as wide a range of ethnic groups as possible. Also, that the data currently available may vary across different ethnic groups. However, T2DM is a complex disease with a combination of numerous genetic and non-genetic factors. In the era of genome-wide association studies (GWAS), where researchers are looking for the genetic causes underlying disease by examining the combined effects of millions of SNPs, a study of two SNPs on 108 T2DM cases and 52 controls samples should investigate the association between polymorphisms (independently of each other and also together) and T2DM using multiple biostatistical methods in addition to comparing allele frequencies.

The result underlying the manuscript in it current form are, in my opinion, not sufficient for publication in the Genes. In any case, I consider it necessary to perform further (more complex) analyses, as the database is available to the Authors.

Author Response

I accept the Authors' answers to points 1 to 6. However, their response to my suggestions is not acceptable to me.

Response 7:

„The authors consider that the Chi-squared test is adequate to illustrate the frequencies of genotypes in the T2DM group and control group for the specific polymorphisms studied, which was our initial aim.”

I respect the Authors' original aim, which was to compare allele frequencies between case and control groups. The Chi2 test used to test this is consistent with this aim. The CHi2 test has limited statistical power and does not correct for potential confounders.

„More complex statistical analyses were not conducted because our aim was not to correlate biochemical parameters (e.g. glucose levels, etc.) with genotypes of specific polymorphisms, but to investigate how the specific SNPs (LEPR Gln223Arg and TNF-α−308G/A) are associated with T2DM in the Greek population, for which limited data exist.”

If the aim was to investigate the association between T2DM and the two SNPs studied, I particularly do not understand why the logistic regression analysis I suggested is not performed. This statistical analysis could confirm that, beyond the difference in allele frequencies, there is an age- and sex-independent (and any other covariate used) association between the SNPs and T2DM.

Furthermore, if it is possible to investigate the association between SNPs and T2DM (and the biochemical parameters associated with it) using more complex statistical methods, the new results of which would confirm the previous one obtained from the Chi2 test already presented, then I really do not understand why these analyses are not being done.

The small sample size (n = 160) and non-random sampling (case-control) would also justify the use of additional statistical methods.

„Additionally, according to the recent bibliography (Shoily et al., 2021), this information builds up to an existing valuable data base of appropriate biomarkers for effective future therapies.”

In that article, they used the 1000 Genomes phase 3 release database, which was designed to create a global reference database (ref: Auton, Adam, Gonçalo R. Abecasis, David M. Altshuler, et al. "A Global Reference for Human Genetic Variation." Nature, vol. 526, no. 7571, 2015, pp. 68-74.). The 160 samples collected in the present study would not be considered representative of the general Greek population (which was not the aim of the present study), but therefore not applicable to perform analyses similar to Shoily et al.

Response 8:

Authors believe that data regarding the association between the studied polymorphism TNF-alpha G-308A with the various chronic diseases are conflicting and not well understood. Moreover, these associations in general, seem to vary from country to country as it is pointed out in the study of Shoily et al, 2021, therefore it is important to investigate the frequencies of polymorphisms in specific ethnic groups and geographic areas. In this context, the study that was conducted on Caucasian Greek individuals is novel. The study of Shoily et al. (2021) was added to the discussion.

I agree with the Authors that genetic analysis should be carried out on as wide a range of ethnic groups as possible. Also, that the data currently available may vary across different ethnic groups. However, T2DM is a complex disease with a combination of numerous genetic and non-genetic factors. In the era of genome-wide association studies (GWAS), where researchers are looking for the genetic causes underlying disease by examining the combined effects of millions of SNPs, a study of two SNPs on 108 T2DM cases and 52 controls samples should investigate the association between polymorphisms (independently of each other and also together) and T2DM using multiple biostatistical methods in addition to comparing allele frequencies.

The result underlying the manuscript in it current form are, in my opinion, not sufficient for publication in the Genes. In any case, I consider it necessary to perform further (more complex) analyses, as the database is available to the Authors.

Response:

We would like to thank the Reviewer for the very useful comments. According to the suggestions proposed by the Reviewer, a multinomial logistic regression analysis by SPSS was carried out to investigate the association between SNPs and the independent variables (gender, age and T2DM group). The resulting information of this analysis was added in the Results section of the manuscript.